# Methods for Evaluating the Efficacy of Medical Castration: A Systematic Review

**DOI:** 10.3390/cancers15133479

**Published:** 2023-07-03

**Authors:** Adriana Aguilar, Jacques Planas, Enrique Trilla, Juan Morote

**Affiliations:** 1Department of Urology, Vall d’Hebron Hospital, 08035 Barcelona, Spain; jacques.planas@vallhebron.cat (J.P.); enrique.trilla@vallhebron.cat (E.T.); juan.morote@vallhebron.cat (J.M.); 2Department of Surgery, Universitat Autònoma de Barcelona, 08193 Bellaterra, Spain

**Keywords:** prostate cancer, androgen deprivation therapy, serum testosterone, free testosterone, luteinising hormone

## Abstract

**Simple Summary:**

Medical castration is the most frequent form of androgen deprivation therapy given as primary treatment for hormone-sensitive metastatic prostate cancer (PCa) and as neo-adjuvant treatment in radiation therapy for intermediate and high-risk localised and locally advanced PCa. PCa guidelines recommend determining serum testosterone levels during castration to assess its efficacy and define castration resistance. Some studies suggest other biochemical serum compounds may assess the effectiveness of castration, such as free testosterone and luteinising hormone (LH). This systematic review aims to analyse the current evidence of biochemical candidates to properly assess medical castration efficacy. We found low-quality evidence supporting the current method to measure serum testosterone during medical castration. All reported studies used immunoassays to quantify the serum testosterone levels of men undergoing medical castration instead of the appropriate liquid chromatography with tandem mass spectrometry (LC-MSMS) to measure low testosterone concentrations. Longitudinal studies are needed to analyse the prognostic value of the serum testosterone measured with LC-MSMS and measure serum-free testosterone or LH.

**Abstract:**

Measuring serum testosterone determination during medical castration is recommended by prostate cancer (PCa) guidelines to assess its efficacy and define castration resistance. It has been suggested that other biochemical compounds, such as free testosterone or luteinising hormone (LH), could also assess castration efficacy. We aimed to analyse the current evidence for serum biochemical compounds that could be appropriate candidates for evaluating medical castration efficacy. A systematic review was conducted after two investigators independently searched the literature in the PubMed, Cochrane Library, and EMBASE databases published between January 1980 and February 2023. Their searches used the medical subject headings ‘prostatic neoplasms’, ‘testosterone and androgen antagonists’, ‘gonadotropin-releasing hormone/analogues and derivatives’, ‘free testosterone’, and ‘luteinising hormone’. Studies were selected according to the Preferred Reporting Items for Systematic Reviews and Meta-Analyses criteria, and their eligibility was based on the Participants, Intervention, Comparator, and Outcome strategy. The search was limited to original articles published in English. Among the 6599 initially identified titles, 15 original studies analysing the clinical impact of serum testosterone levels in PCa patients undergoing androgen deprivation therapy (ADT) were selected for evidence acquisition. The risk of bias in individual studies was assessed using the Quality Assessment of Diagnostic Accuracy Studies 2 tool. All selected studies used immunoassays to measure serum testosterone, although only methods based on liquid or gas chromatography and mass spectrometry are recommended to measure low testosterone concentrations. The reported series were not uniform in clinical stage, ADT types, and the time or number of serum testosterone measurements. Only some studies found low serum testosterone levels (<20 or <32 ng/dL) associated with greater survival free of biochemical progression and castration resistance. We conclude that little current evidence justifies the measurement of serum testosterone during ADT using no appropriate methods. No reported longitudinal studies have examined the clinical impact of serum testosterone measured using liquid chromatography with tandem mass spectrometry (LC-MSMS), free testosterone, or LH in PCa patients undergoing medical castration. We conclude that well-designed longitudinal studies examining the clinical impact of serum testosterone measured with LC-MSMS, serum-free testosterone, and LH on biochemical progression and castration resistance in PCa patients undergoing neo-adjuvant castration in radiation therapy or continuous castration are needed.

## 1. Introduction

Prostate cancer (PCa) is the most diagnosed malignant neoplasm among men in industrialised countries and the second leading cause of cancer-related death in the United States [1,2]. Evidence suggests that early detection of clinically significant PCa reduces the incidence of advanced disease and PCa-specific mortality [3]. After the US Preventive Task Force position statement against PCa screening with prostate-specific antigen in 201 [3], the incidence of local disease decreased from 88.1% to 80.5%. However, the incidence of locally advanced PCa increased from 1.1% to 2.6% and metastatic disease from 4.1% to 7.6% [4].

Androgen deprivation therapy (ADT) is the primary treatment for hormone-sensitive metastatic PCa [5] and a neo-adjuvant treatment in radiation therapy, with different lengths of treatment according to intermediate and high-risk localised or locally advanced disease [6]. ADT aims to reduce the dehydro-testosterone concentration in the nucleus of PCa cells by reducing testosterone synthesis from cholesterol in the testicles through castration. It decreases testosterone synthesis using cytochrome P450 family 17 subfamilies A member 1 (CYP17A1/CYP17) inhibitors, antiandrogens inhibiting androgen receptor dimerization, and compounds reducing active testosterone’s internalisation into the nucleus [7]. Luteinising hormone-releasing hormone (LH-RH) analogues are the most frequent form of medical castration, although LH-RH antagonists and surgical castration immediately decrease serum testosterone [8].

The US Food and Drug Administration (FDA) required an assessment of castration efficacy before approving products for castration introduced during the 1980s, mainly LH-RH analogues [9]. The FDA established 50 ng/dL as the castration threshold for serum testosterone based on the lowest concentration observed in PCa patients who underwent bilateral orchidectomy using the available radioimmunoassays (RIAs) at the time [10]. The Prostate Cancer Working Group 2 introduced the term “castration resistance” in 2008 to characterize the occurrence of clinical and/or biochemical progression despite a castration level of testosterone [11]. The introduction of fast, cheap, and automatable chemiluminescent assays (CLIAs) in the late 1990s enabled the spread of serum testosterone measurement in clinical laboratories, and many PCa guidelines began to recommend its measurement to assess castration efficacy [12,13]. In 2007, Morote et al. reported that PCa patients with micro-elevations in serum testosterone >32 ng/dL during LH-RH analogue treatment showed lower survival free of castration resistance than those without them [14]. Other studies have suggested that maintaining serum testosterone levels <20 ng/dL correlated with better outcomes in PCa patients undergoing medical castration [15]. RIAs and CLIAs usually overestimate low testosterone concentrations compared to liquid or gas chromatography with mass spectrometry (LC/GC-MS). Liquid chromatography with tandem mass spectrometry (LC-MSMS) is currently the recommended method for measuring low testosterone concentrations [16,17]. Some studies have suggested that other serum biochemical compounds can assess castration efficacy, such as free testosterone [18] or LH [19,20].

This systematic review aims to analyse the current evidence on the biochemical candidates for properly evaluating medical castration efficacy and prognosis of PCa patients undergoing this treatment.

## 2. Evidence Acquisition

Two independent authors performed a literature search of the PubMed, Cochrane Library, and EMBASE databases using the medical subject headings ‘prostatic neoplasms’, ‘testosterone and androgen antagonists’, ‘gonadotropin-releasing hormone/analogues and derivatives’, ‘free testosterone’, and ‘luteinising hormone’ to identify articles published between January 1980 and February 2023. Following the Preferred Reporting Items for Systematic Reviews and Meta-Analyses (PRISMA) criteria [21], 6599 titles were identified. This systematic review was registered in PROSPERO (International prospective register of systematic reviews), with the ID number CRD42023424528. After screening titles and abstracts, 499 studies conducted on human adults and published in English were selected. Of these studies, 75 analysing serum testosterone determination in patients undergoing ADT were selected. Finally, 15 original articles analysing the clinical impact of serum testosterone levels during ADT were selected for evidence acquisition according to the patient Population, Intervention, Comparison, and Outcomes strategy [22]. The following criteria were established: (i) adult patients with localised, locally advanced, or metastatic PCa undergoing ADT; (ii) serial biochemical serum testosterone determinations; (iii) analysis of the clinical impact on biochemical progression, castration resistance, cancer-specific survival, or overall survival. We limited the search to original articles published in English; review articles, meta-analyses, and letters were excluded. A flowchart of the study selection process is provided in Figure 1.

Using the Quality Assessment of 120 Diagnostic Accuracy Studies 2 (QUADAS-2) tool [23], we evaluated bias in individual studies. The assessment encompassed four domains: patient selection, index test, reference standard, and flow and timing. Two independent investigators assessed the overall bias of these studies, categorizing it as low (+), high (−), or unclear (?). The summarized results of the QUADAS-2 assessment, presented in Table 1 and Figure 2, indicate that all the studies included in our analysis were determined to be of moderate quality.

## 3. Evidence Synthesis

We present this evidence synthesis by first describing the most relevant items of the selected studies in Table 2. Then, we present the evidence synthesis according to the serum compound proposed to assess castration efficacy and the characteristics and results observed in a series of localised and locally advanced PCa (Table 3), metastatic tumours (Table 4), and a series with mixed clinical stages (Table 5).

### 3.1. Serum Total Testosterone Measurement

#### 3.1.1. Localised and Locally Advanced PCa

In 2007, Morote et al. reported a retrospective study analysing micro-elevations in serum testosterone determined by a CLIA (Immulite^®^, DPC Inc., Los Angeles, CA, USA) in 73 patients with non-disseminated PCa treated with LH-RH analogues. Serum testosterone measurements performed at 6, 12, and 18 months after starting ADT were analysed. The primary objective was to evaluate the survival free of castration resistance. The authors found 32 ng/mL to be the minimum threshold for serum testosterone with clinical impact, showing lower survival free of castration resistance in those who presented these micro-elevations. Castration resistance was observed in 62% of participants during a follow-up of 51 months. It was also shown that patients with micro-elevations >50 ng/dL benefited from a maximal androgen blockade adding bicalutamide, providing a similar rate of castration resistance-free survival to patients without micro-elevations [14].

In 2012, Pickles et al. reported a retrospective study analysing 11,752 patients with PCa who underwent primary radiation therapy between 1998 and 2007 in British Columbia, Canada. They finally selected 2196 patients who received neoadjuvant LH-RH analogues for 3 to 12 months with some serum testosterone measurements during ADT treatment. The method to quantify serum testosterone was a manual RIA used until 2004, and later a CLIA (Elecsys^®^, Roche Inc., Mannheim, Germany). A total of 4954 serum testosterone determinations were analysed, averaging two per patient. This study aimed to examine the association between minor increases in serum testosterone levels and biochemical recurrence based on the Phoenix definition (serum prostate-specific antigen [PSA] nadir + 2 ng/mL). Over an average follow-up period of 45 months, the group with no micro-elevations exceeding 20 ng/dL demonstrated a biochemical relapse rate of 73%, while the group with micro-elevations exceeding 30 ng/dL had a relapse rate of 68.4%. In comparison, the group with micro-elevations surpassing 50 ng/dL exhibited a lower relapse rate of 57.6% (*p* < 0.001) [25].

In 2015, Klotz et al. retrospectively analysed the serum testosterone levels determined during the first year of ADT in 626 patients with non-disseminated PCa who underwent continuous treatment with LH-RH analogues. This study represents a sub-analysis conducted within the control group of the PR-7 study. The primary objective of the PR-7 study was to compare the effectiveness of intermittent and continuous ADT in patients experiencing biochemical recurrence following radical prostatectomy, with or without adjuvant or salvage radiotherapy. The study started in Canada in 1998 and later expanded to include participants from the US and UK. The patients included in this sub-analysis could receive adjuvant or salvage radiotherapy within 12 months before their inclusion. Additionally, they exhibited a serum prostate-specific antigen (PSA) level of 3 ng/mL, no indications of metastatic spread, a serum testosterone level above 144 ng/dL and had undergone at least three serum testosterone measurements during the first year of follow-up. The method used to measure serum testosterone in each centre was not specified. This sub-analysis aimed to evaluate the association of serum testosterone levels with castration resistance-free, specific, and overall survival. Castration resistance was defined as three consecutive increases in serum PSA level, separated by an interval of more than one month, reaching >4 ng/mL, and a serum testosterone level <87 ng/dL. The minimum, mean, and maximum serum testosterone levels were divided into <20 ng/dL, 20–50 ng/dL, and >50 ng/dL. Castration resistance events were detected in 226 participants (37%). The median time to castration resistance was ten years, and the five-year castration resistance-free survival rate was 69%. Castration resistance-free survival was ten years in patients with a serum testosterone nadir of <20 ng/dL, 7.2 years in patients with a serum testosterone nadir of 20–50 ng/dL, and 3.6 years in patients with a serum testosterone nadir of >50 ng/dL (*p* < 0.001). Significant differences in serum testosterone were also associated with specific survival. Mean serum testosterone levels differed significantly with castration resistance-free survival but not specific survival. Finally, the median castration resistance-free survival was not reached when the maximum serum testosterone level was <20 ng/dL. Survival free of castration resistance was 8.9 years in patients with maximum serum testosterone levels of 20–50 ng/dL and ten years in patients with maximum serum testosterone levels of >50 ng/dL (*p* < 0.001). Significant differences in specific survival were also found. The median castration resistance-free survival of patients with mean serum testosterone levels of >50 ng/dL was 4.2 years, lower than the 6.4 years of patients with mean serum testosterone levels of 20–50 ng/dL. Moreover, it did not reach 50% in those patients with mean serum testosterone levels of <20 ng/dL (*p* < 0.01). Serum testosterone nadir values only differed significantly between patients with serum testosterone levels <20 ng/dL and >50 ng/dL. Finally, no significant differences were found in maximum serum testosterone levels. Neither analysis showed significant differences in specific survival [15].

In 2017, Tombal et al. reported a posthoc analysis of the ICELAND trial, a large European study demonstrating the efficacy of continuous administration of leuprorelin (Eligard^®^). This study aimed to examine the association between serum testosterone levels, measured within the first year of continuous treatment, and survival outcomes and time to progression in patients with locally advanced or relapsing non-metastatic PCa. The measurement of serum testosterone was performed using the Elecsys^®^ electro-chemiluminescent immunoassay from Roche Inc., Mannheim, Germany. The patients enrolled in the study were randomly assigned in a 1:1 ratio to receive either continuous (n = 361) or intermittent (n = 340) ADT with leuprorelin over a duration of 36 months. For patients receiving continuous ADT, their minimum, median, and maximum testosterone levels during the first year of therapy were used to stratify them into subsets based on testosterone levels: ≤20, 20–50, and >50 ng/dL. The study findings revealed no significant differences in survival free of PSA progression and cause-specific survival among the subsets established based on serum testosterone levels [32].

In a study conducted by Ozyigit et al. in 2019, the impact of different castrate testosterone levels (<50 and <20 ng/dL) on biochemical relapse-free survival (BRFS) was assessed in patients with non-metastatic intermediate and high-risk PCa who underwent definitive radiation therapy and ADT. The analysis included 173 patients with intermediate- and high-risk PCa, with a median age of 69, who were treated between April 1998 and February 2011. ADT was administered as total androgen blockade using LH-RH analogues and an antiandrogen. The measurement of serum testosterone levels was performed using CLIA, with the Immulite^®^ 2000 system from Siemens Inc., United States. All patients received three months of neoadjuvant ADT followed by radiation therapy and an additional six months of ADT. The biochemical relapse was defined according to the American Society of Radiation Oncology (ASTRO) Phoenix definition. The median follow-up period was 125 months. Among the patients, 96 (56%) had castrate serum testosterone levels below 20 ng/dL, while 139 (80%) had below 50 ng/dL. Patients with castrate serum testosterone levels below 20 ng/dL exhibited 5- and 10-year BRFS rates of 90% and 83%, respectively (*p* = 0.001). Patients with castrate serum testosterone levels below 50 ng/dL demonstrated 5- and 10-year BRFS rates of 86% and 76%, respectively (*p* = 0.006). Therefore, both cutoffs were found to be valid in predicting BRFS. However, patients with castrate serum testosterone levels below 20 ng/dL had significantly better BRFS than other groups (*p* = 0.003). When comparing the two cutoffs, 20 ng/dL was a better predictor of BRFS than 50 ng/dL [35].

In 2021, Tremblay et al. analyzed patients from the continuous ADT arm of the PR-7 trial. Inspired by the findings of Klotz et al. [15], who established that nadir serum testosterone levels during the first year of ADT could predict survival free of castration resistance, this study aimed to determine the comparative significance of serum testosterone levels and serum PSA levels during ADT. Additionally, the study examined whether the occurrence of multiple micro-elevations (>50 ng/dL) in serum testosterone levels could predict castration resistance, cancer-specific survival, and overall survival. However, the specific method used to measure serum testosterone was not specified. Overall, the study concluded that while serum testosterone levels had some prognostic value, their significance in predicting outcomes was overshadowed by the prognostic value of concurrent serum PSA levels. The presence of serum testosterone levels exceeding 20 ng/dL during the first year of ADT was linked to subsequent increases exceeding 50 ng/dL. However, the number of micro-elevations in serum testosterone per patient was not associated with the risk of castration resistance, cancer-specific survival, or overall survival. As expected, a time-dependent adjusted analysis indicated that serum PSA levels were prognostic, whereas cumulative exposure to testosterone did not significantly affect the outcomes [36].

#### 3.1.2. Metastatic PCa

In 2009, Perachino et al. conducted a retrospective study involving 129 patients with disseminated PCa who received the LH-RH analogue goserelin. The study involved quarterly measurements of serum testosterone levels using a chemiluminescent immunoassay (CLIA) method with the Immulite^®^ system from DPC Inc., Los Angeles, CA, USA. The primary objective of the study was to evaluate specific survival. The average follow-up period lasted 47 months, during which 71 events (55%) were observed. Logistic regression analysis revealed that the Gleason score, serum PSA level, and serum testosterone level at the sixth month of treatment emerged as independent predictors of specific survival [24].

In 2015, Yasuda et al. conducted a retrospective analysis on 69 patients with metastatic PCa who underwent maximum androgen blockade using LH-RH analogues and bicalutamide between 2004 and 2010. The measurement of serum testosterone levels was performed using a chemiluminescent immunoassay (CLIA) method with the Elecsys^®^ system from Roche Inc., Mannheim, Germany. These measurements were taken every 3–6 months during the follow-up period until biochemical relapse occurred. On average, 5.5 measurements were taken per patient. During a mean follow-up period of 40 months, 62 patients (90%) experienced castration resistance, and 20 patients (30%) died from a specific cause. However, no significant association was found between serum testosterone levels and survival free of castration resistance or specific survival [28].

In 2016, Shiota et al. conducted a retrospective study involving 96 Japanese patients with metastatic PCa treated between 1996 and 2012. The study aimed to explore the relationships between serum testosterone levels during ADT, body mass index (BMI), castration resistance-free survival and overall survival. Additionally, the study examined the association between a specific gene polymorphism (rs523349) in the steroid five alpha-reductase 2 (SRD5A2) gene and serum testosterone levels during ADT in a subset of 86 cases. Serum testosterone levels were measured randomly using an electrochemiluminescence immunoassay, typically performed twice per patient. All patients received primary treatment through surgical or medical castration using an LH-RH analogue with or without an antiandrogen. Castration resistance was defined based on several criteria, including an increase in PSA > 2 ng/mL and a 25% increase over the lowest recorded level, the emergence of new lesions, or the progression of one or more existing lesions as classified by the Response Evaluation Criteria in Solid Tumors. For patients who experienced disease progression, the overall survival from progression was defined as the period starting from the occurrence of progression to either death or censoring. When serum testosterone levels during ADT were divided into quartiles, no significant differences were observed in castration resistance-free survival or overall survival among the quartiles. However, the lowest quartile tended towards better overall survival and better survival from castration resistance. The study did not find any association between BMI and prognosis. Regarding the SRD5A2 gene polymorphism, the CC allele, which encodes the less active 5-alpha reductase, was found to be associated with lower serum testosterone levels during ADT [30].

Finally, Wang et al. 2017 conducted a study involving 206 patients with metastatic PCa who were followed between January 2007 and September 2012. The ADT regimen consisted of LH-RH analogues and bicalutamide. For patients who experienced biochemical progression, secondary hormonal therapy involved LH-RH agonist administration and withdrawal of bicalutamide for six weeks. Serum testosterone levels were measured before, at one, three, and six months after initiating maximal androgen blockade using an automated CLIA (Access^®^, Beckman Coulter Inc., Fullerton, CA, USA). The primary endpoint of the study was the time from the start of maximal ADT to the development of castration resistance, as indicated by an increase in PSA levels. After one month of ADT, serum testosterone levels were categorized as follows: ≤25 ng/dL in 98 patients (47.6%), 25–50 ng/dL in 95 patients (46.1%), and ≥50 ng/dL in 13 patients (6.3%). The results of both univariate and multivariate analyses showed no significant association between testosterone levels below 50 ng/dL and the effective duration of ADT. However, it was observed that testosterone levels of ≤25 ng/dL after the first month of ADT demonstrated the highest sensitivity and specificity in predicting a longer time until the development of castration resistance (*p* = 0.013) [31].

#### 3.1.3. Localised, Locally Advanced and Metastatic PCa

In 2013, Dason et al. conducted a prospective study involving 32 patients with localised PCa treated with LH-RH analogues or antagonists. Serum testosterone levels were measured every three months using an automated CLIA (Advia-Centaur^®^, Siemens Inc., Tarrytown, NY, USA). The study aimed to assess the impact of serum testosterone levels on castration resistance-free survival. It is worth noting that 7 out of the initially included 39 patients (17.9%) were excluded due to having serum testosterone levels exceeding 50 ng/dL. The researchers evaluated cutoffs of 20 and 32 ng/dL and the mean testosterone levels during the first year of ADT at six and nine months of treatment. The study included 14 patients with disseminated PCa, five with locally advanced PCa, and 13 with biochemical recurrence after local treatment. The average follow-up period was 26 months; half of the patients’ experienced events. Castration resistance was defined based on two consecutive increases in PSA levels above the nadir value. The results showed that a mean serum testosterone level below 32 ng/dL was associated with a longer duration of castration resistance-free survival. This association was also observed when evaluating serum testosterone levels at nine months of ADT but not at six months. However, a cutoff below 20 ng/dL during the first year of ADT did not show a significant association with castration resistance-free survival [26].

In 2013, Bertaglia et al. conducted a prospective study involving 153 PCa patients treated with LH-RH analogues between 2002 and 2006. Serum testosterone levels were measured using a CLIA (Architect^®^, Abbott Inc., Lake Forest, IL, USA) at six and seven months of follow-up. To ensure consistency, the lower measurement from these two-time points was chosen for the analysis. Among the participants, 53 had metastatic or locally advanced PCa, while 99 had biochemical progression after local treatment. The average follow-up period was 65 months. During this period, there were 72 progressions (47%), defined as either a PSA increase of over 50% from the nadir value or the emergence of new radiological lesions.

Additionally, 51 deaths (33%) were recorded. The addition of bicalutamide to LH-RH analogues was performed in 59 out of the 69 patients with micro-elevations in serum testosterone levels above 50 ng/dL. Multivariate analysis showed no significant association between serum testosterone levels and castration resistance-free survival or specific survival. However, when considering serum testosterone levels below 20, below 30, and 50 ng/dL, it was observed that serum testosterone levels below 30 and 50 ng/dL were associated with better specific survival but not with disease progression. Considering the thresholds of 20, 30, and 50 ng/dL, the actuarial analysis did not show any significant differences among patients with biochemical recurrence. However, notable differences were observed in disseminated or locally advanced patients. These findings were consistent with the observations in patients who experienced serum testosterone micro-elevations above 50 ng/dL and received bicalutamide [27].

In 2015, Kamada et al. conducted a retrospective analysis involving 225 patients with PCa treated between 1999 and 2015. The treatment approach included maximal androgen blockade using LH-RH analogues (94%), orchidectomy (3.6%), degarelix (3.1%), bicalutamide (93%), or flutamide (7%). Among the patients, 104 had metastatic PCa, 51 had locally advanced PCa, and 70 had biochemical recurrence after local treatment. Serum testosterone levels were measured at three-month intervals using a CLIA (Architect^®^, Abbott Inc. Lake Forest, IL, USA). The follow-up period was not specified, but 28 deaths (12%) were recorded during this time, and 52 patients (23%) received docetaxel, although the number of castration resistance episodes was not provided. No significant association between castration resistance-free survival and serum testosterone levels at the sixth month of treatment or the nadir testosterone levels was found. However, they did observe that patients with a serum testosterone nadir below 20 ng/dL exhibited better overall survival [29].

In 2017, Sayyid et al. conducted a retrospective study involving 514 PCa patients with different disease stages (localised, locally advanced, or metastatic) treated with continuous ADT. The study focused on those patients with serum testosterone levels below 20 ng/dL, measured using a CLIA (Architect^®^, Abbott Inc., Lake Forest, IL, USA) at the University Health Network between 2007 and 2016. Patients in the study were monitored from the initial measurement of their serum testosterone level below 20 ng/dL until they reached castration resistance, experienced death, or the study concluded. The main focus of the study was to analyze the incidence of serum testosterone elevations above certain thresholds (>20, >30, and >50 ng/dL) and their association with the development of castration resistance. The median duration of follow-up for the patients was 20.3 months. The study found that within five years of follow-up, 82% of patients experienced serum testosterone levels above 20 ng/dL, 45% above 30 ng/dL, and 18% above 50/ng/dL. However, it was noted that 96% to 100% of these patients subsequently re-established serum testosterone levels below 20 ng/dL within the same five-year period. Interestingly, the occurrence of testosterone elevations above these thresholds was not found to be a significant predictor of progression to castration resistance [33].

In 2017, Yamamoto et al. conducted an analysis involving 222 patients with advanced PCa and clinical stage T ≥ 3. These patients were treated with ADT at Chiba University Hospital from 1999 to 2016. Serum testosterone levels were measured using a chemiluminescent immunoassay (CLIA) with the Architect^®^ system by Abbott Inc., Lake Forest, IL, USA. The study aimed to evaluate the prognostic significance of serum testosterone levels and other clinical factors in relation to PSA levels, both in terms of progression-free survival during first-line therapy and overall survival. The mean duration of follow-up was 60.5 months. The median baseline testosterone level was 482 ng/dL, and the mean nadir serum testosterone level was 13 ng/dL. None of the variables associated with serum testosterone were found to be predictive of progression-free survival. However, the multivariate analysis determined that a nadir testosterone level > 20 ng/dL and a reduction in serum testosterone >480 ng/dL were independent prognostic factors for overall survival. The multivariate analysis also identified nadir PSA < 0.1 ng/mL, lymph node metastases, and time to nadir PSA as independent prognostic factors for progression-free survival [34].

This systematic review should acknowledge the inclusion of the study on Relugolix. Relugolix, an oral LH-RH antagonist recently approved, has not yet been the subject of any reported study investigating the prognostic significance of serum testosterone for castration-resistant PCa. However, in a comparative analysis, it was found that a higher proportion of men who received relugolix maintained serum testosterone levels (measured via LC-MSMS) below 50 ng/dL for 48 weeks (96.7%) compared to men receiving leuprolide (88.8%), with a significant difference (*p* < 0.001). Furthermore, a significantly greater percentage of men achieved castrate levels of serum testosterone on day 4 of treatment with relugolix (56%) compared to leuprolide (0%) (*p* < 0.001). Similarly, the cumulative rate of men with serum testosterone levels below 20 ng/dL on day 15 of treatment was significantly higher with relugolix (78.4%) compared to leuprolide (1.0%) (*p* < 0.001). These findings highlight the superior ability of relugolix to maintain lower and more rapidly achieved serum testosterone levels compared to leuprolide in the given timeframes [37].

### 3.2. Serum-Free Testosterone Measurement

No published longitudinal studies have examined correlations between serum levels of free testosterone during treatment and survival free of biochemical progression in PCa patients undergoing neo-adjuvant ADT to radiotherapy or survival free of castration resistance in patients undergoing continuous ADT alone.

### 3.3. Serum LH Measurement

Similar to serum-free testosterone, no published longitudinal studies have examined correlations between serum LH levels and BRFS in patients undergoing neo-adjuvant ADT to radiotherapy or survival free of castration resistance in patients undergoing continuous ADT alone.

## 4. Discussion

ADT is the primary treatment for hormone-sensitive metastatic PCa [5] and neo-adjuvant treatment in radiation therapy for intermediate- and high-risk localised or locally advanced PCa. Medical castration is currently the most frequent form of ADT [6]. The American Urological Association/ASTRO/Society of Urologic Oncology PCa guidelines indicate a serum testosterone threshold of 50 ng/dL for castration [38]. The European Association of Urology PCa guidelines also establish a serum testosterone threshold of 50 ng/dL for castration. However, they noted that patients with serum testosterone levels < 20 ng/dL during treatment usually have better outcomes than those with <20 ng/dL [39].

The evidence obtained in this systematic review based on 15 studies suggests no global consensus on the serum testosterone threshold during ADT with clinical impact on survival free of biochemical progression or castration resistance. All studies placed the serum testosterone threshold for castration <50 ng/dL. However, there was variability between studies using <20 ng/dL [15,25,29,31,33,34,35] and ~30 ng/dL [14,26,27]. The heterogeneity of clinical stages included in the reported studies is remarkable. Six studies included patients with localised and locally advanced PCa [14,15,25,32,35,36], four studies only included patients with metastatic PCa [24,28,30,31], and five studies included patients with localised, locally advanced, and metastatic PCa [26,27,29,33,34]. We noted that some studies had men with biochemical progression after primary treatment without an explicit recommendation of ADT, while others included men with non-metastatic PCa undergoing ADT alone, contrary to the current recommendations in PCa guidelines [38,39].

Regarding the ADT type used, maximal androgen blockade associating LH-RH analogues with bicalutamide was reported in two studies [28,29] and medical castration with LH-RH analogues or antagonists in the others. Morote et al. included patients undergoing both ADT modalities [14], and Bertaglia et al. added bicalutamide when micro-elevations >50 ng/dL were detected [27]. Shiota et al. reported surgical castration in addition to LH-RH analogues [30]. The time and number of serum testosterone measurements during ADT varied. There was also variability when analysing the behaviour of serum testosterone levels, with levels dropping below specific thresholds, micro-elevations surpassing specific thresholds, or medians of all measurements assessed. Given the lack of homogeneity among the appropriate treatments, variability in the time and number of serum testosterone measurements was also an important cause of the low evidence obtained in this systematic review. 

In three of six studies analysing localised or locally advanced PCa, serum testosterone levels < 20 ng/dL were associated with better survival free of biochemical progression, castration resistance, or cancer-specific survival [15,25,35]. One of six analysed studies found that the serum testosterone threshold with clinical impact on survival free of castration resistance was 32 ng/dL [14]. In two studies, no differences were found in survival free of castration resistance or cancer-specific survival when analysing different serum testosterone thresholds [32,36]. When reviewing the four studies that included metastatic PCa, only one reported serum testosterone levels < 20 ng/dL associated with increased survival free of castration resistance [31]. Some studies that included localised, locally advanced, and metastatic PCa patients found some serum testosterone thresholds associated with survival free of castration resistance [26,27,29,33,34]. However, we believe that their results are inconsistent due to the influence of the clinical stages on the prognosis.

The main pitfall identified in this systematic review examining the clinical impact of serum testosterone levels during ADT in patients with PCa was the use of different CLIAs among studies. Indeed, the two important and referenced studies analysing the control arm of the PR-7 trial did not specify the method used to measure serum testosterone and provided contradictory results [15,36]. Between 2003 and 2004, two important studies compared the standard LC/GC-MS method with the clinically available RIAs and CLIAs in women and children [16] and adult men [17]. These studies concluded that assessed RIAs and CLIAs did not correlate with the standard testosterone measurement method at low levels. The Endocrine Society stated in 2007 that low serum testosterone levels, such as those observed in children and women, should be measured using LC/GC-MS [40]. LC-MSMS has been recently integrated into automated platforms and introduced in clinical laboratories with restrictions, even though it is more expensive than CLIAs [41]. Existing evidence shows that CLIAs offer a cheap, fast, susceptible, and automatable serum testosterone measurement, they are not accurate and reproducible at low testosterone levels [41].

Furthermore, no correlation between different CLIAs was observed when analysing the low serum testosterone levels of PCa patients undergoing medical castration [42]. This scenario questions the robustness of the analysed studies regarding the prognostic value of the low serum testosterone levels reached during castration [42,43]. We are concerned about all the clinical studies that have used CLIAs to measure serum testosterone levels, particularly since medicine agencies require serum measurements with LC-MSMS to approve products for castration [40]. Unfortunately, no PCa guidelines emphasise the importance of using an appropriate method for measuring serum testosterone levels in patients undergoing castration [39].

Finally, some recent studies have suggested that other serum biochemical compounds could be used to assess castration efficacy, such as serum-free testosterone [18] or serum LH [19,20]. Recently, it has been observed that serum LH was more efficient in assessing the castrate environment than serum testosterone, regardless of the measurement method [19]. However, no reported longitudinal studies show the clinical impact of serum testosterone measured with LC-MSMS, free testosterone, or LH on the survival free of biochemical progression in men undergoing neo-adjuvant medical castration to radiation therapy or on castration resistance in metastatic PCa patients undergoing continuous castration.

Furthermore, the serum testosterone levels in patients with PCa who undergo castration and receive novel compounds for ADT undergo alterations. Specifically, when abiraterone acetate is administered in combination with castration (as observed in the LATITUDE trial [44]), the serum testosterone levels are lower compared to castration alone. Conversely, using new antiandrogens like bicalutamide in combination therapy may result in a slight increase in serum testosterone levels [45,46,47]. Therefore, relying solely on serum testosterone levels to assess the effectiveness of castration can be challenging, and consideration of other compounds, such as serum LH, may be suggested [19,20]. 

Furthermore, recent findings from the Stampede phase III trial have provided evidence that high-risk non-metastatic PCa patients who underwent radiotherapy along with ADT and abiraterone plus prednisolone experienced an increased duration of metastasis-free survival compared to those who received ADT alone [48].

It is essential to recognize that the evaluation of castration efficacy in the context of combined ADT with new compounds based on serum testosterone is complex and may not be necessary. The current definition of castration-resistant PCa within standard ADT may be outdated, and a new concept for hormone-resistant PCa must be developed.

Finally, it is noteworthy that Relugolix, an oral LH-RH antagonist, has been recently approved. The serum testosterone levels of male patients with metastatic hormone-sensitive PCa who participated in a phase I trial were examined after receiving the combination of relugolix with either abiraterone or apalutamide. It was observed that when abiraterone was combined with relugolix, the serum testosterone levels were lower compared to when apalutamide was combined with relugolix. Nonetheless, in both combinations, the measured serum testosterone levels remained within the castrate range [49]. However, there is currently no reported study that analyzes the prognostic significance of serum testosterone for castration-resistant PCa [37].

## 5. Conclusions

The heterogenicity of studies analysing the benefit of serum testosterone measurement during ADT in PCa patients is currently extremely high. The main pitfall is the lack of studies measuring the low serum testosterone levels of castrate PCa patients with an appropriate method. While automated LC-MSMS is available at clinical reference laboratories, it is more expensive than the various CLIAs. Therefore, well-designed longitudinal studies using LC-MSMS to measure serum testosterone levels in PCa patients under castration are needed. However, serum-free testosterone and LH should be explored as candidate serum compounds for assessing castration efficacy.

## Figures and Tables

**Figure 1 cancers-15-03479-f001:**
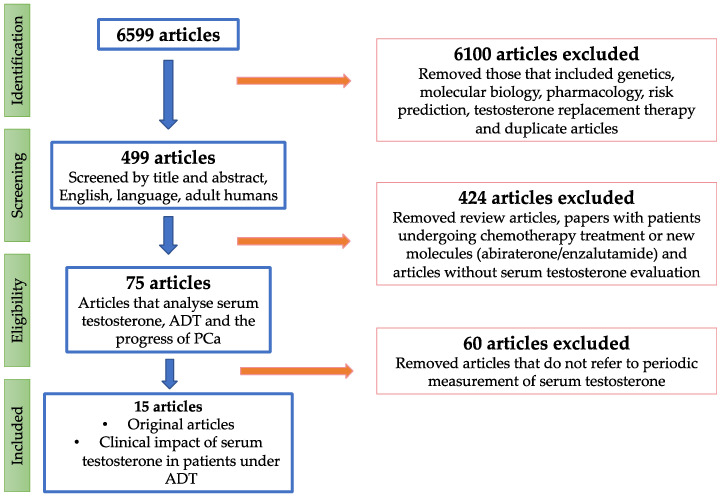
PRISMA flow diagram.

**Figure 2 cancers-15-03479-f002:**
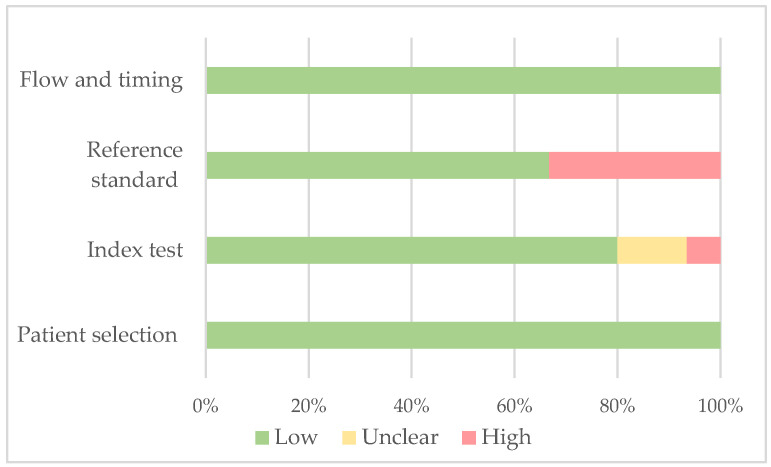
The risk of bias graph about each risk of bias item is presented as percentages across all included series.

**Table 1 cancers-15-03479-t001:** Summary of risk of bias; +: low risk of bias; −: high risk of bias; ?: unclear risk of bias:.

	Patient Selection	Patient Selection	Index Test	Index Test	Reference	Flow and Timing
Author, Year	Selection Criteria Clearly Described	Clinical Data Available	Castration Threshold Pre-Specified	Serum Testosterone Measurement Method Specified	Reference Standard	Adequate Follow-Up
Morote et al., 2007 [14]	+	+	+	+	+	+
Perachino et al., 2009 [24]	+	+	?	+	−	+
Pickles et al., 2012 [25]	+	+	+	+	+	+
Dason et al., 2013 [26]	+	+	+	+	+	+
Bertaglia et al., 2013 [27]	+	+	+	+	+	+
Yasuda et al., 2015 [28]	+	+	?	+	−	+
Klotz et al., 2015 [15]	+	+	+	−	+	+
Kamada et al., 2015 [29]	+	+	+	+	+	+
Shiota et al., 2016 [30]	+	+	?	+	−	+
Wang et al., 2017 [31]	+	+	?	+	+	+
Tombal et al., 2017 [32]	+	+	+	+	−	+
Sayyid et al., 2017 [33]	+	+	+	+	+	+
Yamamoto et al., 2017 [34]	+	+	+	+	+	+
Ozyigit et al., 2019 [35]	+	+	+	+	+	+
Tremblay et al., 2021 [36]	+	+	+	−	−	+

**Table 2 cancers-15-03479-t002:** Summary of global characteristics of the studies that have analysed the influence of serum testosterone levels on the evolution of patients with PCa undergoing ADT.

Author, Year	nº Patients	Clinical Stage	Treatment Received	Follow-Up Method	Time of Determination (Months)	Measurement Method
Morote et al., 2007 [14]	73	50 LA; 23 BR	45 LH-RH; 28 MAB	ST	6, 12, 18	CLIA
Perachino et al., 2009 [24]	129	M	LH-RH	ST	every 3	CLIA
Pickles et al., 2012 [25]	2196	L and LA	RT and LH-RH	ST	serial measurements (mean 2 months)	CLIA
Dason et al., 2013 [26]	32	14 M; 5 LA; 13 BR	LH-RH and LHRH-ant	ST	every 3	CLIA
Bertaglia et al., 2013 [27]	153	51 M; 99 BR	LH-RH	ST	at 6	CLIA
Yasuda et al., 2015 [28]	69	M	MAB	ST	every 3–6	CLIA
Klotz et al., 2015 [15]	626	BR	LH-RH agonist	ST	every 2 (1 y)	non reported
Kamada et al., 2015 [29]	225	70 L; 51 LA; 104 M	MAB	ST	every 3	CLIA
Shiota et al., 2016 [30]	96	M	9 LH-RH, 87 LH-RH + surgical castration	ST	2 times (1–5)	CLIA
Wang et al., 2017 [31]	206	M	LH-RH	ST	1,3,6	CLIA
Tombal et al., 2017 [32]	361	LA + BR	LH-RH	ST	every 6	CLIA
Sayyid et al., 2017 [33]	950	L + LA + BR + M	LH-RH	ST	every 1–4	CLIA
Yamamoto et al., 2017 [34]	222	LA + M	LH-RH	ST	non reported	CLIA
Ozyigit et al., 2019 [35]	173	L	RT and LH-RH	ST	every 3 (2 y); every 4 (3 and 4 y); every 6 (thereafter)	CLIA
Tremblay et al., 2021 [36]	678	BR	LH-RH	ST	every 2 (2 y)	non reported

LA: locally advanced PCa; M: Metastatic PCa; L: localized PCa; BR: biochemical recurrence; LH-RH: LH-RH agonist; LH-RH-ant: LH-RH antagonist; MAB: maximal androgen blockade; ST: serum testosterone.

**Table 3 cancers-15-03479-t003:** Characteristics of the studies that included localized and locally advanced PCa patients.

Author, Year	PSA ng/mL	Threshold Value	Event	Aim	Summary
Morote et al., 2007 [14]	81.2 (mean)	<32 ng/dL	CR	CRFS	32 ng/dL was the minimum level of ST with an impact on CRFS. If micro elevations > 50 ng/dL, using bicalutamide improved CRFS.
Pickles et al., 2012 [25]	-	<20 ng/dL	ST ↑	CRFS	Microelevations > 30–50 ng/dL predict 58% CRFS at five years. If there are no microdeletions, the CRFS rate increases to 78%.
Klotz et al., 2015 [15]	-	<20 ng/dL	CR	CRFS, CSS	Nadir ST < 20 ng/dL showed higher CRFS and CSS rates. Microelevations >50 ng/dL are linked to lower rates of CRFS and CSS.
Tombal et al., 2017 [32]	-	X	CR	CRFS, CSS	No differences in CSS and CRFS progression among the ≤20 ng/dL, >20 to ≤50 ng/dL, and >50 ng/dL testosterone-level subgroups.
Ozyigit et al., 2019 [35]	14 (median)	<20 ng/dL	BR	BRFS	Both <20 ng/mL and <50 ng/mL ST levels are valid for predicting BRFS. However, <20 ng/dL have significantly better BRFS compared to <50 ng/mL
Tremblay et al., 2021 [36]	-	X	CR	CRFS, CSS, OS	ST > 20 ng/dL in the 1st year of ADT was observed to result in increases with rises >50 ng/dL. The number of ST breakthroughs had no association with CRPC, CSS or OS.

CR: castration resistance; ST ↑: serum testosterone micro elevations. CRFS: castration resistance-free survival; CSS: cancer-specific survival; BRFS: biochemical relapse-free survival; OS: overall survival.

**Table 4 cancers-15-03479-t004:** Characteristics of the studies that included metastatic PCa patients.

Author, Year	PSA ng/mL	Threshold Value	Event	Aim	Summary
Perachino et al., 2009 [24]	185.8 (mean)	-	Death	CSS	Lower levels of ST at 6 m after ADT, higher CSS
Yasuda et al., 2015 [28]	610 (mean)	-	CR	CRFS and OS	No prognostic impact in CRFS and OS of ST during MAB
Shiota et al., 2016 [30]	181.8 (mean)	-	CR	CRFS and OS	The lowest quartile of serum testosterone levels during ADT was a significant predictor of better OS and CRFS.
Wang et al., 2017 [31]	241 (median)	<25 ng/dL	CR	CRFS	ST ≤ 25 ng/dL after 1 m of ADT: best CRFS. ST 25 ng/dL after 1 m of ADT can distinguish patients who may benefit from adding docetaxel.

**Table 5 cancers-15-03479-t005:** Characteristics of the studies included localised, locally advanced, and metastatic PCa patients.

Author, Year	PSA ng/mL	Threshold Value	Event	Aim	Summary
Dason et al., 2013 [26]	70.8 (mean)	<32 ng/dL	CR	CRFS	Microelevations >50 ng/mL excluded. ST at 9 m < 32 ng/dL, better CRFS. Mean ST during the 1st year of ADT < 32 ng/dL better CRFS.
Bertaglia et al., 2013 [27]	21 (mean)	<30 ng/dL	CR	CRFS and OS	Differences only in M patients. ST at 6 m < 20 ng/dL higher OS. ST at 6 m < 20 vs. 20–50 vs. >50 ng/dL is related to CRFS.
Kamada et al., 2015 [29]	42.6 (mean)	<20 ng/dL	CR	CRFS and OS	Nadir ST during follow-up <20 ng/dL higher OS but not improves CRFS.
Sayyid et al., 2017 [33]	19.04 (median)	<20 ng/dL	CR	CRFS	Continuous ADT with initial ST < 20 ng/dL showed substantial long-term variations in ST, and lacked prognostic significance in CRPC progression.
Yamamoto et al., 2017 [34]	86.03 (median) 571.5 (mean)	<20 ng/dL	CR	CRFS and OS	Testosterone reduction >480 ng/dL and ST < 20 ng/dL and are prognostic factors for primary ADT in advanced PCa.

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
