# Peer review of "Methods for Evaluating the Efficacy of Medical Castration: A Systematic Review"

_cancers, 2023, doi:10.3390/cancers15133479_

Round 1
Reviewer 1 Report
The systematic review presented here analyses papers that address the effect of drug castration on achieved testosterone levels.
The methodology, tabulation and description of the content of each of the 15 papers included in the final analysis is very good.
The following points need to be addressed in the results and/or discussion:
Modern antiandrogens such as abiraterone, enzalutamide, apalutamide and darolutamide are not mentioned, although they have a massive impact on testosterone levels by mechanism of action and, moreover, can achieve a massive survival benefit in both the hormone-sensitive metastatic situation and the castration-resistant metastatic situation.
The combination of the testosterone receptor antagonist bicalutamide with an LHRH analogue is described, but is no longer appropriate in view of the many drugs mentioned above (See Terrain study). However, bicalutamide is the only approved testosterone receptor antagonist that can be administered as monotherapy (without castration).
Abiraterone leads to castration via Zyp-17 inhibition, so in combination with the LHRH analogue, supercastration must result, which is probably responsible for the improved oncological outcomes. However, I am not aware of any papers describing the presumed particular depth of testosterone levels with this combination.
The new testosterone receptor antagonists enza-, apa-, and darolutamide block intra- and extracellular testosterone receptors. Given as a monosubstance, they should even lead to an increase in testosterone, like bicalutamide. Given in combination, they will not lower serum testosterone, but will ensure that residual testosterone, no matter how low, can no longer have an oncologic effect.
All guidelines recommend today the addition of one of the above-mentioned new drugs in every metastasised situation, if necessary also as a so-called triplet in combination with chemotherapy. In this respect, the testosterone depth achieved by the LHRH analogue alone is actually no longer very relevant in metastases, because either the addition of abiraterone ensures supercastration or an effect of the residual testosterone is effectively prevented (enza-, apa-, darolutamide).
Only in neoadjuvant and adjuvant radiation therapy for localised Pca is ADT still given as monotherapy. In this context, however, the achieved depth of testosterone is not very important, because this concomitant and time-limited therapy does not achieve a survival benefit anyway, but only a biochemical progression advantage.
Relugolix is an oral LHRH antagonist that has been approved in the USA since 2022 and in Europe since 2023. Its effectiveness needs to be included in the analysis.
Reviewer 2 Report
The authors recommend the use of LC-MSMS in measuring blood testosterone levels, based on previous reports. This is an interesting review.
1. There are two different units of blood testosterone concentration, ng/dl and nmol/L, which is difficult to understand. Both converted values should be stated.
2. It would be interesting that the LH-like substance, hCG, should be discussed.
Round 2
Reviewer 1 Report
Points 1, 2, 3, and 5 were well addressed.
Points 4 and 6: Agreement in the response letter does not imply a substantial improvement to the manuscript. Please add appropriate sentences and/or modify the existing text.
Point 7: Relugolix is approved for the treatment of advanced hormone-sensitive prostate cancer. The FDA & EMA approval text does not mention the mandatory presence of metastases or castration resistance. The authors' argument for why it was not evaluated is therefore incomprehensible. This substance must be included in the comparative analysis.
Round 3
Reviewer 1 Report
All comments were taken and appropriate additions made to the manuscript.